# Short-duration phase-change material cryotherapy selectively enhances early neuromuscular and perceptual recovery

Chong Gang[1☯], Hao Wang[1☯], Yujue Wang[2], Yongsheng Lan[1]*

1 Jilin Provincial Key Laboratory for the Enhancement of Athletic Performance (Comprehensive Physical Fitness Monitoring), Changchun Normal University, Changchun, Jilin, China, 2 Faculty of Sports and Exercise Science, University Malaya, Kuala Lumpur, Malaysia

☯ These authors contributed equally to this work.
* Lanys@ccsfu.edu.cn

## Abstract

The purpose of this study was to examine whether short-duration phase-change material cooling (PCM) applied at different temperatures influences acute recovery following fatigue induced by stretch–shortening cycle exercise (SSC). Sixty-four physically active participants were randomly assigned to 5°C, 10°C, or 15°C PCM cryotherapy group or a passive recovery control group. After completing a SSC fatigue protocol, participants underwent a 15-minute PCM intervention, and peak torque (PT), mean power, rate of force development (RFD), countermovement jump (CMJ) performance, Rating of perceived exertion (RPE), modified endurance ratio (MER), vastus lateralis (VL) and Rectus Femoris (RF) stiffness were assessed immediately after fatigue (Imm-fatigue), immediately after PCM cryotherapy (Imm-PCM), and 60 minutes post PCM cryotherapy (Post60-PCM). Mean power and RFD were significantly greater in PCM groups compared with the control group at Imm-PCM ($P \le 0.01$), with mean power remaining elevated in the 15°C PCM group at post-60-PCM ($P \le 0.05$). RPE was significantly lower in all PCM groups at Imm-PCM and post60-PCM compared with control ($P \le 0.01$). No between-group differences were observed for PT, CMJ, MER, or muscle stiffness, and no temperature-dependent effects were detected within the 5–15 °C. These findings indicate that 15-minute PCM cryotherapy selectively accelerates early-phase neuromuscular and perceptual recovery without affecting maximal strength, endurance capacity, or passive muscle mechanical properties. From an applied perspective, PCM cryotherapy may be an effective strategy to enhance explosive performance and perceived readiness during short recovery intervals in training or competition settings.

**Data availability statement:** All relevant data are within the manuscript.

**Funding:** The funder (Grant of Jilin Provincial Education Department, Grant number: JJKH20261248KJ) had a role in study design, data collection and analysis, and the decision to publish.

**Competing interests:** The authors have declared that no competing interests exist.

## Introduction

In high-intensity competitive sports, athletes are frequently required to perform multiple bouts within a short time frame, leaving limited opportunity for physiological recovery. Repeated eccentric–concentric activities, such as those involved in sprinting or countermovement jumps (CMJ), can rapidly induce neuromuscular fatigue characterized by reduced muscle strength, power, and rate of force development, accompanied by elevated perceived exertion and muscle stiffness [1]. These symptoms result from metabolic disturbances, including phosphocreatine (PCr) depletion, accumulation of hydrogen ions ($H^+$) and inorganic phosphate (Pi), and transient impairment of excitation–contraction coupling [2]. Without sufficient recovery, such fatigue can compromise subsequent performance and increase the risk of musculoskeletal injury [3]. Therefore, identifying efficient strategies to accelerate recovery between successive performances is of critical importance in both training and competition contexts [4].

Cryotherapy has long been used as a recovery intervention aimed at reducing tissue temperature to modulate local blood flow, metabolic rate, and inflammatory activity [5]. Physiologically, cooling decreases metabolic demand, reduces membrane permeability, and suppresses nociceptive transmission through reduced nerve conduction velocity, thereby alleviating pain and delaying secondary muscle damage [6]. Recent advancements in cryotherapy have introduced phase change material (PCM) technology, which provides prolonged and stable cooling through latent heat exchange at a constant preset temperature. Unlike traditional methods such as cold-water immersion or ice packs, PCM cooling offers advantages of temperature precision, comfort, and portability, allowing its practical application during competition intervals [7].

However, the optimal temperature range and acute physiological effects remain insufficiently characterized [8]. Previous studies have suggested that within the so-called "safe therapeutic zone" (5–15°C), reductions in tissue metabolic rate tend to plateau, implying a potential thermal threshold effect [9]. Furthermore, most existing research has focused on long-duration cooling interventions [7,8], whereas short-term PCM applications—which are more relevant for real competition settings—have received limited empirical investigation. It remains unclear how different PCM temperatures acutely influence neuromuscular recovery, perceptual fatigue, and muscle mechanical properties following fatigue induced by stretch–shortening cycle (SSC) exercise [10].

This study employed a randomized controlled design to investigate the acute recovery effects of PCM cryotherapy at different temperatures (5°C, 10°C, and 15°C) following an SSC-induced fatigue model. By evaluating neuromuscular performance (peak torque, mean power, rate of force development, and countermovement jump height), perceptual fatigue, endurance recovery (modified endurance ratio), and muscle mechanical properties (quadriceps stiffness), the study aimed to verify the validity of the fatigue model, determine the short-term recovery efficiency of PCM cooling, and examine whether temperature gradients elicit distinct recovery responses or reflect a physiological threshold effect within the 5–15°C range. The findings are

expected to provide physiological evidence and practical guidance for optimizing short-term recovery strategies in competitive sports.

## Methods

### Subjects

A total of 64 healthy participants (32 males and 32 females, aged: $20.19 \pm 1.33$) majoring in sports training were recruited through convenience sampling. Inclusion criteria were: (1) absence of acute or chronic lower-limb injuries; (2) regular engagement in moderate-to-vigorous exercise; (3) ability to tolerate cold exposure; and (4) provision of informed consent. Participants with muscle or joint disorders, cardiovascular abnormalities, or cold hypersensitivity were excluded. The recruitment period was from 28/October/2024–31/December/2024.

Males and females participants were randomly assigned numbers (1–4) separately using Excel's RANDBETWEEN function, corresponding to the following 4 groups (n = 16 each): control (Con), 5°C, 10°C, and 15°C PCM cryotherapy. The assignment order and group codes were securely locked in a box by a secretary. All participants were fully informed of the study procedures, potential risks, and benefits, and written informed consent was obtained prior to participation.

### Procedures

This study adopted a randomized, single-blind, controlled design to examine the acute recovery effects of PCM cryotherapy following an SSC-induced fatigue protocol. The experimental protocol consisted of 4 phases: Baseline measurement, fatigue induction, intervention, and recovery (Fig 1). The four intervention groups differed only in the temperature of PCM cooling (5°C, 10°C, 15°C) or absence of cooling (control). Cooling lasted 15 min immediately after the fatigue protocol, followed by a 60 min passive recovery period.

### Blinding procedures

The researcher responsible for all neuromuscular performance, perceptual, and muscle mechanical property measurements was blinded to group allocation. To maintain blinding, this researcher remained in a separate testing area during the intervention period and entered the laboratory only when participants were prepared for post-intervention assessments, with no interaction occurring during the PCM application.

### Fatigue protocol

A validated stretch–shortening cycle (SSC) fatigue model was employed, consisting of ten sets of ten maximal counter-movement jumps (CMJ) with 30 s rest between sets. Participants performed all jumps with hands on hips to minimize upper-body contribution. The success of fatigue induction was confirmed by significant post-exercise reductions in PT, RFD, and CMJ height, and increases in Borg RPE and quadriceps stiffness.

### PCM cryotherapy intervention and ethics statement

Commercially manufactured PCM cooling packs (20 × 30 cm) were used, preset to maintain phase-change temperatures of 5°C, 10°C, or 15°C. Before each trial, surface temperature stability was verified using an infrared thermometer (variation ≤ 0.4°C). During PCM Cryotherapy Intervention, two packs were applied to the anterior thigh of each leg and secured with elastic straps while participants remained seated. Cooling duration was standardized at 15 min, and participants refrained from additional physical activity during and after the intervention. This study was approved by Changchun Normal University ethics committee (2024030).

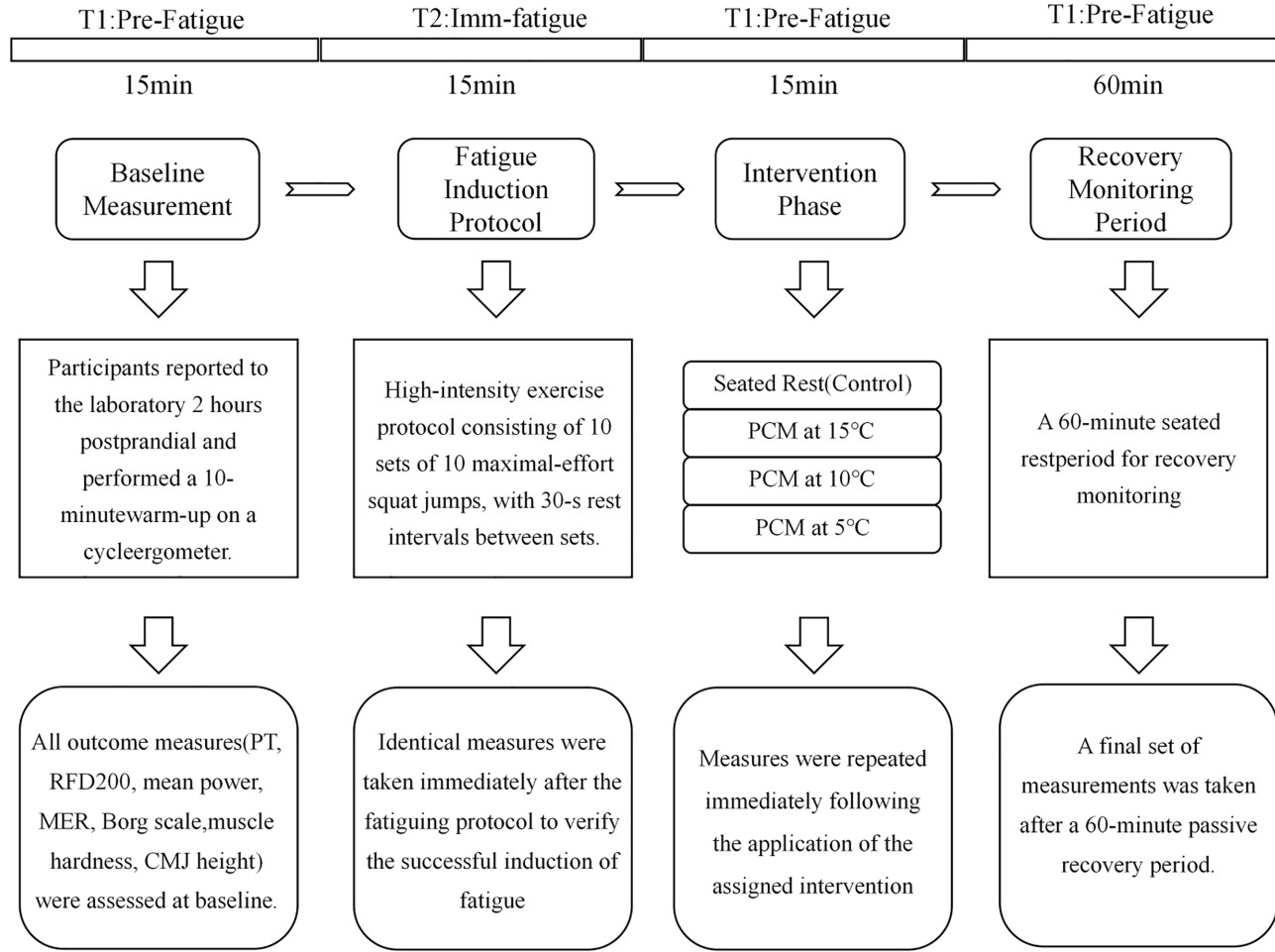

**Fig 1. Overview of experimental design and measurement timeline.**

### PT, MER, RFD, mean power, RPE, CMJ, muscle stiffness measurements

PT, MER, RFD, and mean power were assessed using an isokinetic dynamometer (PHYSIOMED CON-TREX, Germany). All measurements were performed in a concentric–passive knee extension mode, with a total of 20 consecutive repetitions at an angular velocity of $60°·s^{-1}$. Participants were seated on the dynamometer chair with the hip joint fixed at 85°. The length of the distal tibial pad on the lever arm was adjusted to position the contact point approximately 2–3 cm proximal to the lateral malleolus of the tested limb. To minimize extraneous movement, the participant's torso, pelvis, and mid-thigh were secured using stabilization straps. Participants were instructed to cross their arms over the chest with hands placed on the shoulders to prevent upper-limb involvement during testing. Prior to each trial, the rotational axis of the dynamometer was carefully aligned with the anatomical axis of the knee joint. Throughout the test, standardized verbal encouragement and real-time visual feedback (display of joint torque output) were provided to ensure maximal effort. The range of motion for knee extension was set at 70°, corresponding to a movement from 80° to 10° of knee flexion (with 0° defined as full knee extension).

RPE was assessed using the Borg 6–20 scale. Test–retest reliability of the scale was evaluated in a subset of participants (n = 64) who completed the assessment under identical pre-fatigue conditions on two occasions separated

by a two-week interval. Reliability was quantified using the intraclass correlation coefficient calculated with a two-way mixed-effects model for absolute agreement. The Borg 6–20 scale demonstrated excellent reliability, with an intraclass correlation coefficient of 0.870 for single measures (95%CI: 0.795–0.919) and 0.931 for average measures (95% CI: 0.886–0.958).

CMJ height was recorded using a contact timing mat (SmartJump, Fusion Sport, AUS). During the test, the participants should place their hands on their hips, stand with their legs shoulder-width apart, then quickly squat down (bending their hips and knees by 90°) before jumping back up.

Muscle stiffness of the vastus lateralis (VL) and rectus femoris (RF) was measured using a hand-held myotonometer (MyotonPRO, Estonia). To ensure consistency in the measurement locations, the measurement point for the Vastus Lateralis was determined at approximately one-third of the distance between the greater trochanter and the lateral femoral epicondyle. The measurement point for the Rectus Femoris was located at one-third of the distance between the anterior superior iliac spine and the patella.

All indicators were measured at four time points: i) Pre-fatigue (Baseline); ii) immediately post-fatigue (Imm-fatigue); iii) Immediately after PCM cooling (Imm-PCM); iv) 60 min after PCM cooling (Post60-PCM).

## Statistical analysis

Data were analyzed using SPSS 27.0 (IBM, USA). Descriptive statistics were reported as mean ± standard deviation (SD). Fatigue induction was verified by paired t-tests comparing pre- and post-fatigue values. Recovery effects were analyzed using two-way repeated-measures ANOVA, with group (control, 5°C, 10°C, 15°C) as the between-subject factor and time (Imm-fatigue, Imm-PCM, Post60-PCM) as the within-subject factor. When significant interactions were found, post hoc pairwise comparisons with Tukey correction were applied. The significance level was set at $p < 0.05$.

## Results

A total of 64 healthy participants (32 males and 32 females) were enrolled and randomly allocated to 4 groups (n = 16/group): 5°C PCM, 10°C PCM, 15°C PCM, and control. Participant characteristics are summarized in Table 1. No significant between-group differences were observed at baseline for height, weight, BMI, age, PT, MER, RFD, mean power, RPE, CMJ, VL, RF. All data are presented as mean ± standard deviation (SD).

Table 1. Characteristics of participants (n = 64, M ± SD, F:M = 8:8 for each).

|  | Con | 15°C PCM | 10°C PCM | 5°C PCM |
|---|---|---|---|---|
| Height (cm, n = 16) | 173.19 ± 7.88 | 173.69 ± 6.67 | 174.63 ± 9.33 | 172.52 ± 7.85 |
| Weight (kg, n = 16) | 64.08 ± 13.57 | 66.74 ± 8.63 | 68.84 ± 7.02 | 61.64 ± 8.33 |
| BMI (kg/m$^2$, n = 16) | 21.26 ± 3.43 | 22.16 ± 2.92 | 22.65 ± 2.57 | 20.64 ± 1.79 |
| Age (year, n = 16) | 20.88 ± 1.78 | 20.31 ± 1.35 | 19.75 ± 0.93 | 19.81 ± 0.83 |
| PT (Nm·kg$^{-1}$, n = 16) | 2.94 ± 0.33 | 2.88 ± 0.70 | 2.90 ± 0.42 | 2.92 ± 0.53 |
| MER (n = 16) | 0.77 ± 0.08 | 0.77 ± 0.06 | 0.76 ± 0.07 | 0.77 ± 0.07 |
| RFD (Nm·s$^{-1}$, n = 16) | 751.19 ± 186.51 | 794.44 ± 202.84 | 743.34 ± 126.61 | 722.97 ± 212.52 |
| Mean power (W, n = 16) | 96.88 ± 11.70 | 99.69 ± 25.70 | 97.22 ± 18.03 | 93.80 ± 14.14 |
| RPE (n = 16) | 12.19 ± 0.75 | 11.88 ± 1.15 | 12.31 ± 1.08 | 12.25 ± 1.39 |
| CMJ (cm, n = 16) | 36.92 ± 7.34 | 36.99 ± 9.21 | 37.72 ± 8.41 | 37.69 ± 9.14 |
| VL (N·m$^{-1}$, n = 16) | 236.31 ± 22.73 | 246.88 ± 48.70 | 237.12 ± 21.48 | 227.38 ± 16.77 |
| RF (N·m$^{-1}$, n = 16) | 264.69 ± 26.44 | 267.19 ± 20.35 | 273.94 ± 33.37 | 268.38 ± 23.37 |

## Validation of the fatigue model

Paired t-tests confirmed that the stretch–shortening cycle (SSC) fatigue protocol successfully induced a fatigue model (Table 2). Compared with pre-fatigue, PT, MER, RFD, mean power and CMJ decreased significantly (p < 0.001), whereas RPE, VL and RF increased markedly (p < 0.001).

## Neuromuscular functional recovery

For PT, the two-way repeated-measures ANOVA revealed a significant main effect of time (F (2,120) = 120.0, p < 0.001), a significant effect of group × time interaction (F (6,120) = 2.538, p = 0.02). However, no significant main effect of groups (F

**Table 2. Changes in performance and physiological indicators before and after fatigue protocol.**

|  | Mean ± Standard Deviation | |
| --- | --- | --- |
|  | Pre-Fatigue | Imm-Fatigue |
| CMJ-Con | 36.92 ± 7.34 | 26.09 ± 7.87*** |
| PT-Con | 2.94 ± 0.33 | 2.17 ± 0.32*** |
| Borg-Con | 12.19 ± 0.75 | 16.81 ± 1.05*** |
| Mean Power-Con | 96.88 ± 11.70 | 62.74 ± 9.98*** |
| RFD-Con | 751.19 ± 186.51 | 475.34 ± 131.93*** |
| MER-Con | 0.77 ± 0.08 | 0.66 ± 0.07*** |
| VL stiffness-Con | 236.31 ± 22.73 | 296.44 ± 25.96*** |
| RF stiffness-Con | 264.69 ± 26.44 | 305.31 ± 32.54*** |
| CMJ-15°C | 36.99 ± 9.21 | 27.51 ± 8.61*** |
| PT-15°C | 2.88 ± 0.70 | 2.16 ± 0.60*** |
| RPE-15°C | 11.88 ± 1.15 | 16.69 ± 1.40*** |
| Mean Power-15°C | 99.69 ± 25.70 | 71.42 ± 19.09*** |
| RFD-15°C | 794.44 ± 202.84 | 530.62 ± 150.27*** |
| MER-15°C | 0.77 ± 0.06 | 0.65 ± 0.09*** |
| VL stiffness-15°C | 246.88 ± 48.70 | 300.19 ± 40.88*** |
| RF stiffness-15°C | 267.19 ± 20.35 | 322.00 ± 36.39*** |
| CMJ-10°C | 37.72 ± 8.41 | 29.03 ± 6.86*** |
| PT-10°C | 2.90 ± 0.42 | 2.23 ± 0.53*** |
| RPE-10°C | 12.31 ± 1.08 | 17.25 ± 1.00*** |
| Mean Power-10°C | 97.22 ± 18.03 | 69.51 ± 15.70*** |
| RFD-10°C | 743.34 ± 126.61 | 520.06 ± 139.55*** |
| MER-10°C | 0.76 ± 0.07 | 0.66 ± 0.12** |
| VL stiffness-10°C | 237.12 ± 21.48 | 297.38 ± 27.45*** |
| RF stiffness-10°C | 273.94 ± 33.37 | 317.81 ± 39.02*** |
| CMJ-5°C | 37.69 ± 9.14 | 29.01 ± 6.69*** |
| PT-5°C | 2.92 ± 0.53 | 2.28 ± 0.53*** |
| RPE--5°C | 12.25 ± 1.39 | 17.06 ± 0.93*** |
| Mean Power-5°C | 93.80 ± 14.14 | 66.71 ± 15.52*** |
| RFD-5°C | 722.97 ± 212.52 | 539.97 ± 176.90** |
| MER-5°C | 0.77 ± 0.07 | 0.68 ± 0.07** |
| VL stiffness-5°C | 227.38 ± 16.77 | 284.50 ± 18.70*** |
| RF stiffness-5°C | 268.38 ± 23.37 | 312.38 ± 34.89*** |

Compared with Pre-Fatigue, ** P < 0.01; *** P < 0.001.

(3, 60) =1.504, p=0.22), suggesting that 15-min PCM cryotherapy did not confer additional benefits to maximal voluntary torque production beyond those attributable to natural recovery within the first 60 minutes post PCM (Table 3).

For mean power, the two-way repeated-measures ANOVA revealed significant effects of time (F (2,120) = 73.38, p<0.001, Fig 2A), group (F (3,60) = 3.485, p=0.02, Fig 2B), and a group×time interaction (F (6,120) = 2.409, p=0.03). Given the significant main effect of groups, post-hoc analyses (Tukey's test) were performed. At Imm-PCM, the 10°C and 15°C PCM groups exhibited significantly higher mean power than the control group (p<0.01), whereas the 5°C group did not differ from control. By Post60-PCM, only the 15°C group maintained a significantly greater mean power compared with control (p<0.05, Fig 2B). These findings indicate that short-duration PCM cooling can acutely enhance mean power output, with effects most pronounced immediately following intervention and attenuating within 60 minutes.

For RFD, the two-way repeated-measures ANOVA showed a significant main effect of time (F (2,120) = 63.39, p<0.001, Fig 3A), groups (F (3,60) = 2.765, p=0.05, Fig 3B), and group×time interaction (F (6,120) = 2.583, p=0.02). Given the significant main effect of groups, post-hoc analyses (Tukey's test) were performed. At Imm-PCM, all three PCM groups (5°C, 10°C, and 15°C) produced significantly greater RFD values compared with the control group (p<0.05 to p<0.001), indicating a clear acute neuromuscular advantage of cooling. However, these differences disappeared by Post60-PCM, suggesting that the neural and contractile benefits of PCM cooling on rapid force production are transient and confined to the immediate post-intervention phase.

For CMJ, the two-way repeated-measures ANOVA displayed a significant main effect of time (F (2,120) = 154.7, p<0.001, Table 4). However, neither the group effect (F (3,60) = 0.732, p=0.54) nor the group×time interaction (F (6,120) = 1.916, p=0.08) reached statistical significance.

For RPE, the two-way repeated-measures ANOVA exhibited significant main effects of time (F (2,120) = 366.6, p<0.001, Fig 4A), groups (F (3,60) = 8.675, p<0.001, Fig 4B), and group×time interaction (F (6,120) = 9.850, p<0.001). Given the significant main effect of groups, post-hoc analyses (Tukey's test) were performed. At both Imm-PCM and Post60-PCM, all the three PCM groups reported significantly lower RPE than the control group (p<0.001), demonstrating that PCM cryotherapy elicited a pronounced and sustained reduction in subjective fatigue.

For MER, the two-way repeated-measures ANOVA showed a significant main effect of time (F (2,120) = 31.21, p<0.001). However, neither the group effect (F (3,60) = 0.611, p=0.61) nor the group×time interaction (F (6,120) = 1.050, p=0.40) reached significance (Table 5).

## Muscle mechanical properties

For VL and RF, the two-way repeated-measures ANOVA demonstrated a significant main effect of time (F (2,120) = 85.11, p<0.001; F (2,120) = 94.20, p<0.001). However, no significant group effect (F (3,60) = 1.090, p=0.36; F (3,60) = 0.945, p=0.42) or group×time interaction (F (6,120) = 0.174, p=0.98; F (6,120) = 0.517, p=0.79) was detected (Tables 6 and 7).

Table 3. PT across time points for each group (n=16, F:M=8:8 for each, Nm·kg⁻¹).

| | Imm-fatigue | Imm-PCM | Post60-PCM | F test | |
|---|---|---|---|---|---|
| | M±SD | M±SD | M±SD | F | P |
| 15°C | 1.74±0.42 | 2.41±0.49*** | 2.44±0.56*** | | |
| 10°C | 1.66±0.39 | 2.31±0.35*** | 2.36±0.51*** | | |
| 5°C | 1.78±0.36 | 2.20±0.39*** | 2.40±0.39*** | | |
| Control | 1.66±0.35 | 1.94±0.36** | 2.15±0.56*** | | |
| Times | | | | 120 | <0.001 |
| Groups | | | | 1.50 | 0.22 |
| Times*Groups | | | | 2.54 | 0.02 |

Compared with Imm-fatigue **P<0.01 ***P<0.001.

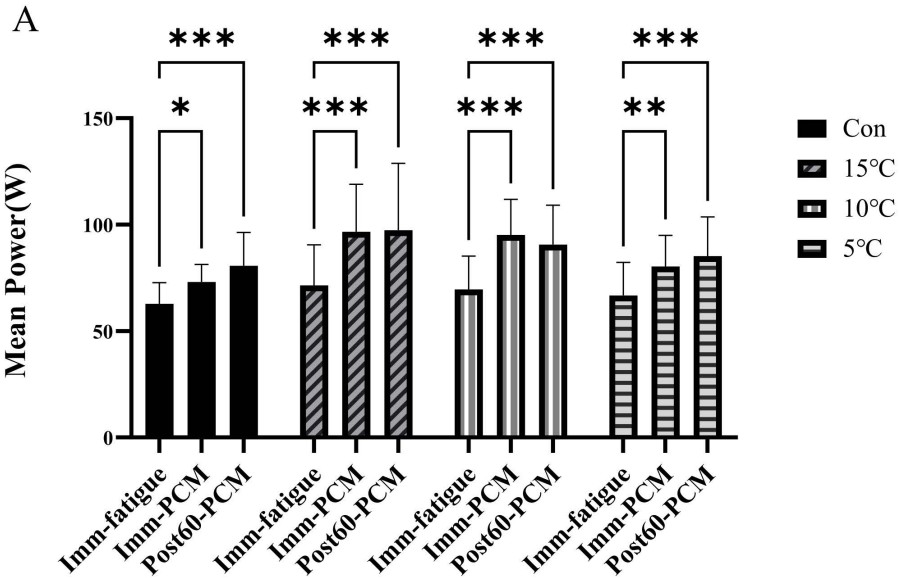

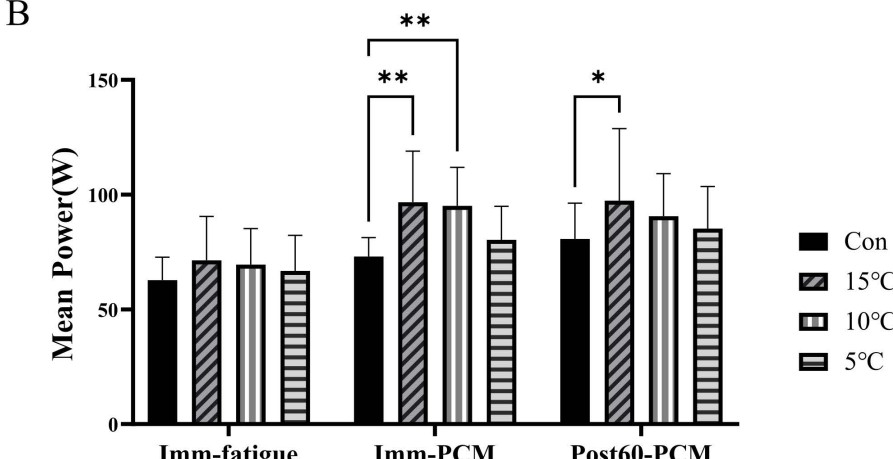

**Fig 2. The effect of temperature and time on mean power in the CON group and PCM – treated groups. A.** Mean power (W) in the CON group (control) and PCM – treated groups under 5°C, 10°C, and 15°C conditions at Imm – fatigue, Imm – PCM, and Post60 – PCM time points. Data are presented as mean ± SD (n = 16). Statistical significance was determined by two-way ANOVA followed by Tukey's post – hoc test: *$p < 0.05$, **$p < 0.01$, ***$p < 0.001$ vs. CON group at the same time point. **B.** Mean power (W) in the CON group and PCM – treated groups under different temperature conditions at Imm – fatigue, Imm – PCM, and Post60 – PCM time points. Data are presented as mean ± SD (n = 16). Statistical significance was determined by two-way ANOVA followed by Tukey's post – hoc test: *$p < 0.05$, **$p < 0.01$.

## Discussion

Collectively, 15 min of PCM cooling selectively enhanced early-phase neuromuscular performance, as evidenced by transient improvements in mean power and rate of force development immediately after intervention, and consistently reduced perceived exertion up to 60 min post-cooling. In contrast, no additional benefits were observed for peak torque, countermovement jump performance, endurance-related capacity, or muscle stiffness, and no temperature-dependent differences were detected within the 5–15°C range. Collectively, these results indicate that PCM cryotherapy primarily facilitates rapid neural and perceptual recovery rather than structural or endurance-related restoration.

A

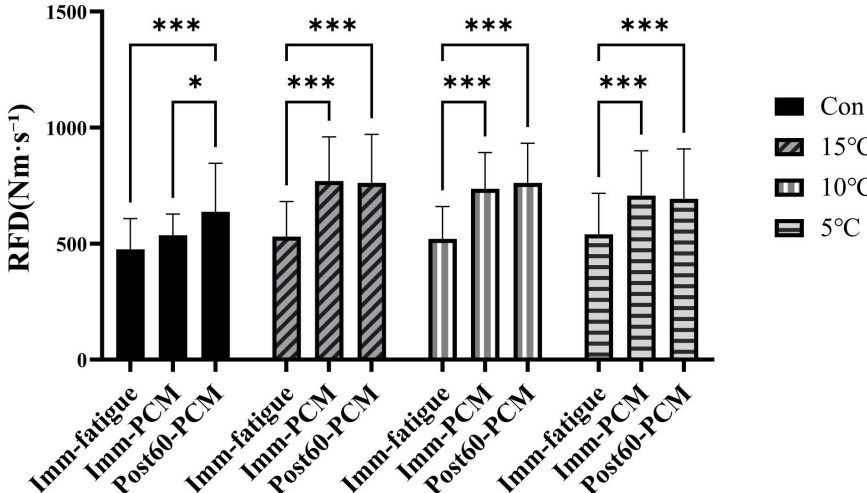

B

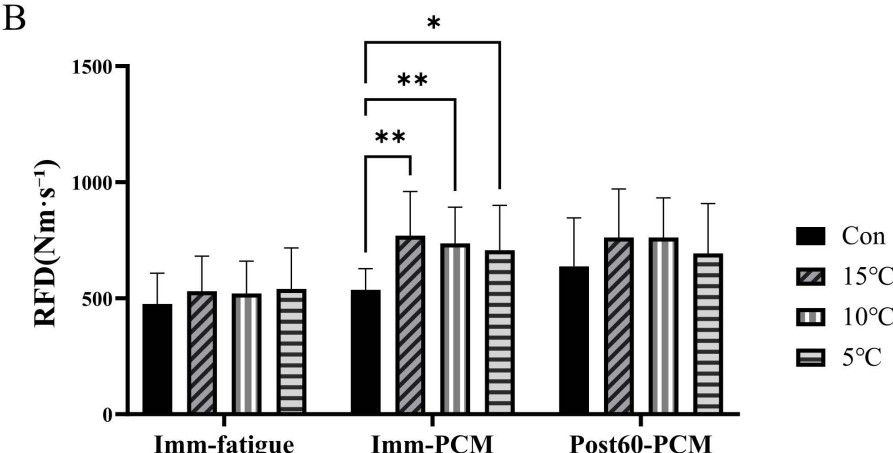

**Fig 3. Rate of Force Development (RFD) responses across time points and intervention groups. A.** RFD values for each group (CON, 15°C, 10°C, 5°C) measured at Imm-fatigue, Imm-PCM, and Post60-PCM. A significant main effect of time and time×group interaction was observed. All PCM conditions (5°C, 10°C, 15°C) demonstrated significantly higher RFD at Imm-PCM compared to CON. **B.** Group-wise comparisons illustrating that RFD was significantly elevated in the 5°C, 10°C, and 15°C PCM groups relative to CON at Imm-PCM. Data are presented as mean±SD. *p<0.05, **p<0.01, ***p<0.001.

The present study demonstrates that short-duration PCM cryotherapy selectively enhances acute neuromuscular recovery, particularly reflected in improvements in mean power and rate of force development (RFD) at the Imm-PCM time point [11]. These findings support the notion that cooling primarily modulates early-phase metabolic and neural fatigue mechanisms, rather than structural muscular recovery [12,13]. The transient elevation in mean power following 10°C and 15°C cooling likely reflects improved ATP-CP turnover efficiency, reduced metabolic by-product accumulation, and partial restoration of excitation–contraction coupling efficiency—all changes associated with temperature-mediated metabolic suppression [12]. By lowering local metabolic demand and attenuating the accumulation of hydrogen ions and inorganic phosphate, cooling creates a more favorable environment for rapid cross-bridge reattachment and force generation during

**Table 4. CMJ across time points for each group (n = 16, F:M = 8:8 for each, cm).**

| | Imm-fatigue | Imm-PCM | Post60-PCM | F test | |
|---|---|---|---|---|---|
| | M±SD | M±SD | M±SD | F | P |
| 15°C | 27.51±8.61 | 32.79±10.08*** | 33.86±10.40*** | | |
| 10°C | 29.03±6.86 | 34.65±8.10*** | 35.65±7.73*** | | |
| 5°C | 29.01±6.69 | 33.72±8.48*** | 34.09±8.55*** | | |
| Control | 26.09±7.87 | 29.17±7.41*** | 32.12±6.53*** | | |
| Times | | | | 154.7 | <0.001 |
| Groups | | | | 0.73 | 0.54 |
| Times*Groups | | | | 1.92 | 0.08 |

Compared with Imm-fatigue ***P < 0.001.

dynamic contractions [14]. RFD, a neural-driven indicator, showed the most robust response to cryotherapy, with all PCM temperatures significantly outperforming the control at Imm-PCM. This enhancement aligns with the recognized sensitivity of RFD to reductions in spinal-level inhibitory input and improvements in motoneuron excitability [15]. Cooling reduces afferent feedback from metabolite-sensitive group III/IV fibers, thereby diminishing inhibitory interneuron drive and enabling faster motor unit recruitment and synchronization [16]. However, both mean power and RFD benefits dissipated by Post60-PCM, indicating that these neuromuscular advantages are time-limited and dependent on the acute neural facilitation and metabolic stabilization that occur only during the immediate post-cooling phase. By contrast, peak torque (PT) and counter-movement jump height (CMJ) did not exhibit significant group differences. PT is primarily influenced by structural muscular properties, including contractile protein integrity, calcium handling, and mechanical cross-bridge capacity—all of which require longer recovery periods than the 15-minute cooling application used here [17]. Similarly, CMJ performance reflects a complex integration of strength, coordination, stretch–shortening cycle (SSC) efficiency, and neuromuscular timing [18]. Although RFD improved, this local neural facilitation was insufficient to modify the broader multi-factor biomechanical demands of CMJ performance within the short recovery window. Together, these observations suggest that PCM cryotherapy accelerates functional, but not structural neuromuscular recovery following SSC-induced fatigue.

The marked reductions in rating of perceived exertion (RPE) at both Imm-PCM and Post 60-PCM indicate that PCM cryotherapy exerts a potent influence on perceptual and central aspects of recovery. Cold exposure is known to suppress nociceptive transmission by reducing nerve conduction velocity and dampening the activity of pain-sensitive receptors [19]. This results in attenuated sensations of soreness, muscle burn, and fatigue, thereby lowering perceived exertion independent of mechanical restoration [20] Additionally, cold-induced shifts in autonomic balance—characterized by reduced sympathetic drive and enhanced parasympathetic activation—can further contribute to the subjective sense of recovery and relaxation [21]. These mechanisms help explain why PCM cryotherapy produced sustained reductions in RPE even when neuromuscular variables such as mean power and RFD no longer differed from control at Post60-PCM.

In contrast, the modified endurance ratio (MER) did not show group-specific effects. MER reflects fatigue resistance and the ability to maintain force over time, which depends heavily on oxidative metabolism, microvascular perfusion, and mitochondrial recovery—processes that unfold over a much longer temporal scale than the acute cooling applied here [22]. Although perceived exertion decreased, this central and sensory improvement did not translate into measurable changes in endurance-related muscular performance. This dissociation underscores an important consideration for practitioners: perceptual recovery may outpace physiological recovery, potentially creating a mismatch between how "ready" an athlete feels and their actual metabolic restoration.

Despite significant reductions in muscle stiffness over time, no between-group differences emerged for VL or RF stiffness across the three PCM temperatures and the control condition. This pattern reflects the inherently slow, structurally

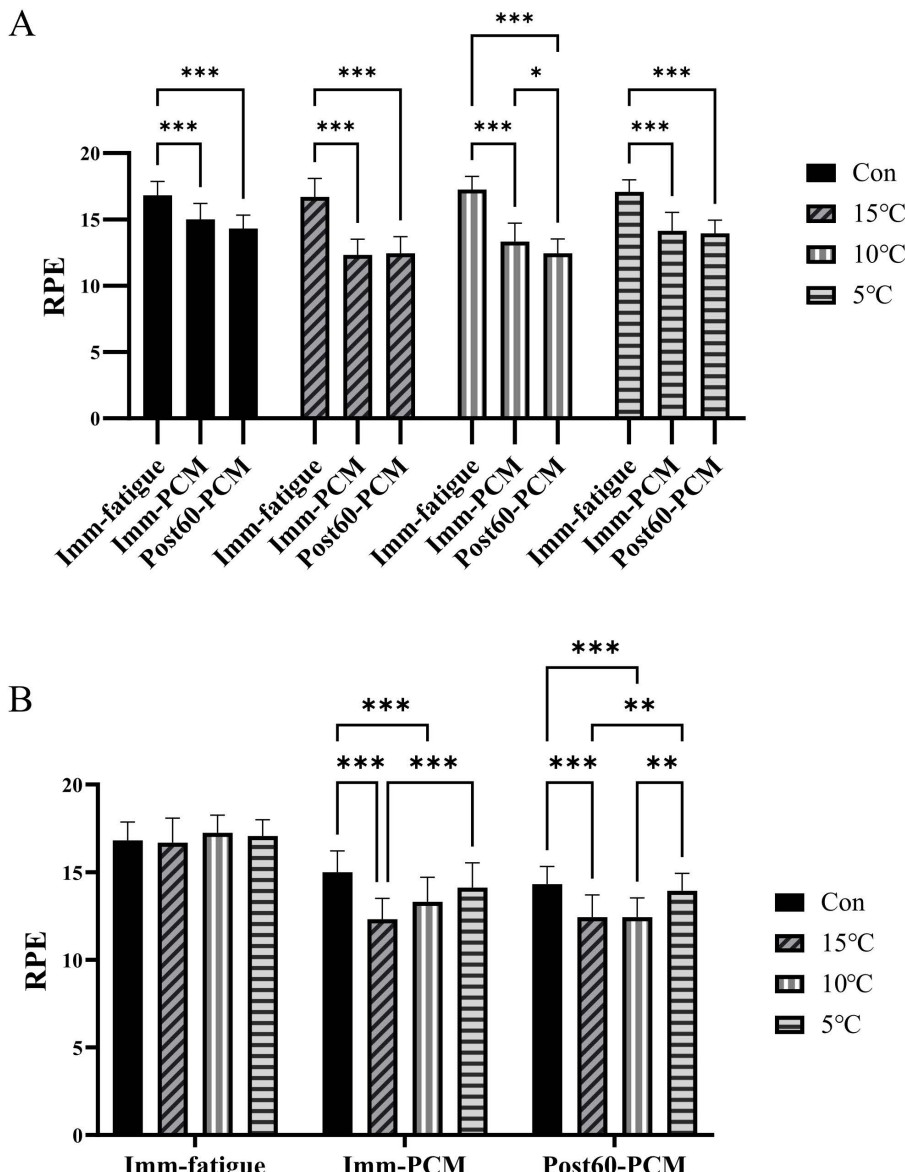

**Fig 4. Borg Rating of Perceived Exertion (RPE) across groups and time points. A.** RPE responses at Imm-fatigue, Imm-PCM, and Post60-PCM across the control (CON), 15°C, 10°C, and 5°C PCM groups. Significant main effects were observed for group, time, and the group×time interaction. PCM groups demonstrated lower perceived exertion than the control at both Imm-PCM and Post60-PCM. **B.** Direct comparison of RPE values at Imm-fatigue, Imm-PCM, and Post60-PCM across all groups. * $p < 0.05$, ** $p < 0.01$, *** $p < 0.001$.

driven nature of muscle mechanical recovery [23]. Stiffness is predominantly determined by the viscoelastic properties of the extracellular matrix, intramuscular connective tissue tension, fluid redistribution, and inflammation-induced cytoskeletal alterations [24,25]. These factors recover through biochemical remodeling and fluid shifts that require several hours, rather than the brief 15-minute cooling window provided in this study. As such, stiffness recovery is largely insensitive to short-term cooling interventions.

Additionally, the lack of temperature-specific effects supports the existence of a therapeutic cooling threshold within the 5–15°C range. Once tissue temperature falls within the reduced metabolic zone, further reductions offer

**Table 5. MER across time points for each group (n=16, F:M=8:8 for each).**

| | Imm-fatigue | Imm-PCM | Post60-PCM | F test | |
|---|---|---|---|---|---|
| | M±SD | M±SD | M±SD | F | P |
| 15°C | 0.65±0.09 | 0.78±0.15*** | 0.71±0.10* | | |
| 10°C | 0.66±0.12 | 0.78±0.10*** | 0.71±0.08 | | |
| 5°C | 0.68±0.07 | 0.78±0.08*** | 0.76±0.07* | | |
| Control | 0.66±0.07 | 0.73±0.07* | 0.72±0.09 | | |
| Times | | | | 31.21 | 0.001 |
| Groups | | | | 0.61 | 0.61 |
| Times*Groups | | | | 1.05 | 0.40 |

Compared with Imm-fatigue *P<0.05; **P<0.01; ***P<0.001.

**Table 6. VL across time points for each group (n=16, F:M=8:8 for each, N·m⁻¹).**

| | Imm-fatigue | Imm-PCM | Post60-PCM | F test | |
|---|---|---|---|---|---|
| | M±SD(%) | M±SD(%) | M±SD(%) | F | P |
| 15°C | 300.19±40.88 | 242.44±50.04*** | 230.19±38.11*** | | |
| 10°C | 297.38±27.45 | 244.81±71.77*** | 225.00±15.51*** | | |
| 5°C | 284.50±18.70 | 225.81±33.90*** | 218.25±29.42*** | | |
| Control | 296.44±25.96 | 231.56±30.06*** | 224.94±19.75*** | | |
| Times | | | | 85.11 | <0.001 |
| Groups | | | | 1.09 | 0.36 |
| Times*Groups | | | | 0.17 | 0.98 |

Compared with Imm-fatigue *P<0.05; **P<0.01; ***P<0.001.

**Table 7. RF across time points for each group (n=16, F:M=8:8 for each, N·m⁻¹).**

| | Imm-fatigue | Imm-PCM | Post60-PCM | F test | |
|---|---|---|---|---|---|
| | M±SD | M±SD | M±SD | F | P |
| 15°C | 322.00±36.39 | 284.88±34.63*** | 279.19±39.74*** | | |
| 10°C | 317.81±39.02 | 284.56±28.64*** | 279.25±26.57*** | | |
| 5°C | 312.38±34.89 | 271.44±31.37*** | 268.63±32.31*** | | |
| Control | 305.31±32.54 | 275.63±37.44*** | 259.38±34.25*** | | |
| Times | | | | 94.20 | <0.001 |
| Groups | | | | 0.94 | 0.42 |
| Times*Groups | | | | 0.52 | 0.79 |

Compared with Imm-fatigue *P<0.05; **P<0.01; ***P<0.001.

minimal added benefit for mechanical tissue properties [26,27]. This plateau effect aligns with previous evidence indicating that collagen viscoelasticity, microvascular perfusion, and passive muscle tension reach a stabilization point across moderate cooling intensities [28]. Thus, while PCM cryotherapy effectively improves perceptual and functional neuromuscular indices, it does not accelerate early-phase restoration of passive mechanical muscle properties.

## Practical implications

This study demonstrates that a brief (15 min) PCM cryotherapy intervention selectively accelerates early-phase recovery following SSC-induced fatigue by enhancing neural-driven performance indices (mean power and RFD) and reducing perceived exertion, without affecting maximal strength, endurance capacity, or muscle mechanical properties [11,28]. The comparable responses across PCM temperatures (5–15°C) indicate a therapeutic cooling threshold, suggesting that moderate temperatures are sufficient for acute recovery [26,27] Practically, PCM cryotherapy may be effectively implemented during short recovery intervals to improve explosive performance and perceptual readiness, while serving as a complementary strategy rather than a substitute for longer-term recovery processes.

## Limitations and future directions

This study was limited by its relatively homogeneous sample and the absence of biochemical markers that could further elucidate the metabolic mechanisms of recovery. Future research should include diverse athletic populations and combine physiological and molecular indicators to clarify the time–temperature relationship of PCM cryotherapy. Such work will help establish evidence-based guidelines for its practical application in sports recovery.

## Author contributions

**Conceptualization:** Chong Gang, Hao Wang, Yongsheng Lan.

**Investigation:** Chong Gang, Hao Wang, Yongsheng Lan.

**Writing – original draft:** Yujue Wang, Yongsheng Lan.

**Writing – review & editing:** Yujue Wang, Yongsheng Lan.

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
