## [Decision Letter · Decision Letter 0]

18 Mar 2026

Dear Dr. Lan,

Thank you for submitting your manuscript to PLOS ONE. After careful consideration, we feel that it has merit but does not fully meet PLOS ONE’s publication criteria as it currently stands. Therefore, we invite you to submit a revised version of the manuscript that addresses the points raised during the review process.

We look forward to receiving your revised manuscript.

Kind regards,

Julio Alejandro Henriques Castro da Costa

Academic Editor

PLOS One

**Journal Requirements:**

https://journals.plos.org/plosone/s/file?id=wjVg/PLOSOne_formatting_sample_main_body.pdf andandandand

“This article is funded by Grant of Jilin Provincial Education Department (Grant number: JJKH20261248KJ).”

4. Please note that funding information should not appear in any section or other areas of your manuscript. We will only publish funding information present in the Funding Statement section of the online submission form. Please remove any funding-related text from the manuscript.

5. Please amend either the title on the online submission form (via Edit Submission) or the title in the manuscript so that they are identical.

6. Please upload a new copy of Figures 1, 2, 3, and 4 as the detail is not clear. Please follow the link for more information:  https://journals.plos.org/plosone/s/figures

7. We note you have included a table to which you do not refer in the text of your manuscript. Please ensure that you refer to Table 4 in your text; if accepted, production will need this reference to link the reader to the Table.

Reviewers' comments:

Reviewer's Responses to Questions

**Comments to the Author**

1. Is the manuscript technically sound, and do the data support the conclusions?

Reviewer #1: Yes

2. Has the statistical analysis been performed appropriately and rigorously?

Reviewer #1: Yes

3. Have the authors made all data underlying the findings in their manuscript fully available?

Reviewer #1: Yes

4. Is the manuscript presented in an intelligible fashion and written in standard English?

Reviewer #1: Yes

Reviewer #1: Dear Sir/Madame

This is very interesting paper. The simple use of cooloing effect to improve the recovery is a god idea and allready used in top sports - for ex cold water immersion after intensive sports for improving the recovery purpose.

Now there is presented that the physiological parameters will really improve and change with cooling from 5 to 15 degrees - the differences between 5- to 15 degrees differ statistically from controls.

As You also pointed out that further studies will be needed concerning the biochemical markers. That will be also very interesting to know and migth be confirm also the use of these methods as evidence based reasearch.

One thing is if the trial was registerd at Trials.gov register ??

The conclusion was made well. amd also the discussion was well wrote.

.

Reviewer #1: No

---

## [Author Response · Author response to Decision Letter 1]

25 Mar 2026

Thank you for your question. Because this study was designed as a short-term, acute physiological sports science experiment (involving a 15-minute intervention and a 60-minute recovery observation) involving healthy, physically active university students rather than a medical/clinical treatment trial for patients, it was not prospectively registered on ClinicalTrials.gov. However, the study protocol was strictly reviewed and approved by the Ethics Committee of Changchun Normal University (Approval No: 2024030), and all procedures were conducted in full accordance with the Declaration of Helsinki.

---

## [Decision Letter · Decision Letter 1]

9 Apr 2026

Short-Duration Phase-Change Material Cryotherapy Selectively Enhances Early Neuromuscular and Perceptual Recovery

PONE-D-26-01323R1

Dear Dr. Lan,

We’re pleased to inform you that your manuscript has been judged scientifically suitable for publication and will be formally accepted for publication once it meets all outstanding technical requirements.

Kind regards,

Julio Alejandro Henriques Castro da Costa

Academic Editor

PLOS One

Additional Editor Comments (optional):

Reviewers' comments:

Reviewer's Responses to Questions

**Comments to the Author**

Reviewer #1: All comments have been addressed

2. Is the manuscript technically sound, and do the data support the conclusions?

Reviewer #1: Yes

3. Has the statistical analysis been performed appropriately and rigorously?

Reviewer #1: Yes

4. Have the authors made all data underlying the findings in their manuscript fully available?

Reviewer #1: Yes

5. Is the manuscript presented in an intelligible fashion and written in standard English?

Reviewer #1: Yes

Reviewer #1: Authors has responded to my queries propelry. A short term follow-up - so I will agree this answers.

.

Reviewer #1: **Yes:**Olavi Airaksinen, Professor of PRMOlavi Airaksinen, Professor of PRMOlavi Airaksinen, Professor of PRMOlavi Airaksinen, Professor of PRM

---

## [Editor Report · Acceptance letter]

PONE-D-26-01323R1

PLOS One

Dear Dr. Lan,

I'm pleased to inform you that your manuscript has been deemed suitable for publication in PLOS One. Congratulations! Your manuscript is now being handed over to our production team.

Kind regards,

on behalf of

Dr. Julio Alejandro Henriques Castro da Costa

Academic Editor

PLOS One